# The Prevalence of Ocular Extra-Intestinal Manifestations in Adults Inflammatory Bowel Disease: A Systematic Review and Meta-Analysis

**DOI:** 10.3390/ijerph192315683

**Published:** 2022-11-25

**Authors:** Jing-Xing Li, Chun-Chi Chiang, San-Ni Chen, Jane-Ming Lin, Yi-Yu Tsai

**Affiliations:** 1Department of General Medicine, China Medical University Hospital, Taichung 404327, Taiwan; 2School of Medicine, China Medical University, Taichung 404328, Taiwan; 3Graduate Institute of Clinical Laboratory Sciences and Medical Biotechnology, National Taiwan University, Taipei 116216, Taiwan; 4Department of Ophthalmology, China Medical University Hospital, Taichung 404327, Taiwan; 5Department of Optometry, Asia University, Taichung 413305, Taiwan; 6School of Medicine, National Taiwan University, Taipei 116216, Taiwan; 7School of Chinese Medicine, China Medical University, Taichung 404328, Taiwan

**Keywords:** ocular extra-intestinal manifestation, uveitis, episcleritis, conjunctivitis, inflammatory bowel disease, Crohn’s disease, ulcerative colitis

## Abstract

Patients with inflammatory bowel disease (IBD) have a greater frequency of ocular extra-intestinal manifestations (O-EIMs) than the general population, while Crohn’s disease (CD) and ulcerative colitis (UC) have inconsistent prevalence, according to previous studies. This study aimed to examine the prevalence of O-EIMs in CD and UC, respectively. We systemically reviewed O-EIMs and IBD across several online databases. Inclusion criteria are as follows: (1) observational studies examining the association between O-EIMs and IBD, such as cross-sectional, case–control, or cohort studies; (2) human and adult individuals; and (3) with case and control groups consisting of patients with and without O-EIMs, respectively. Patients under the age of 18 or any study on pediatric IBD will be excluded. The prevalence of uveitis in adults was determined by 21 studies comprising 190,941 individuals with IBD, including 62,874 CD and 128,067 UC. The pooled analysis revealed significantly increased odds of uveitis in patients with CD than with UC (pooled odd ratio (OR) 1.603, 95% confidence interval 1.254–2.049). The subgroup analysis revealed that European populations had significantly higher odds of developing uveitis and episcleritis in patients with CD than UC (pooled OR 1.683 and 2.401, respectively). Although O-EIMs may be the prodrome of IBD, no consistent finding was obtained as a result of the high heterogeneity from the two included studies. This meta-analysis indicates the significantly increased odds of uveitis in adults with CD than those with UC. In subgroup analysis, European with CD seemed to have higher odds of uveitis and episcleritis than those with UC. Nonetheless, the link between O-EIMs and IBD remained unclear.

## 1. Introduction

Inflammatory bowel disease (IBD), including Crohn’s disease (CD) and ulcerative colitis (UC), is a chronic, inflammatory, polygenic, multifactorial, and multifactorial illness with unknown etiology. Studies have reported extraintestinal manifestations (EIMs) or issues in several organ systems, including musculoskeletal, dermatologic, ocular, pulmonary, renal, and hepatobiliary systems [1]. Furthermore, the existing literature is inconsistent in terms of whether these EIMs are more associated with CD or UC. Ocular EIMs (O-EIMs) are reported in 4–12% of patients with IBD, and the rate was recorded up to 29% in some cohorts [2]. EIMs are more prevalent in CD than in UC. Ocular symptoms were observed from mild conjunctivitis and episcleritis to severe inflammation, such as scleritis and uveitis. Uveitis is typically more serious and causes vision loss, whereas episcleritis is a benign condition. The most prevalent ocular IBD symptoms are episcleritis (2–5%) and uveitis (0.5–3.5%) [3,4]. Uveitis is a uveal tract inflammation involving the iris, ciliary body, and choroid and may lead to visual impairments [5,6]. According to the predominant site of inflammation, uveitis can be classified as anterior, middle, posterior, or panuveitis. Patients with IBD exhibit bilateral anterior uveitis with a gradual onset [7,8] and are unrelated to IBD activity.

Nonetheless, the prevalence reported by research varies considerably, which makes it difficult to determine if O-EIMs are more prevalent in patients with CD or UC. Although a higher prevalence of O-EIMs has been demonstrated in CD patients compared with UC patients [9], the results are controversial [10,11]. CD and UC represent two separate disease entities. CD can affect any part of the GI tract, but the large and small intestines are most affected, while UC mainly affects the colon and rectum. CD is more severe than UC, yet UC is significantly more prevalent worldwide. Moreover, the risk factors of CD and UC were substantially distinct [12]. 

This study aims to examine the prevalence of O-EIMs in patients with CD and UC and further analyze the disparity in different world regions. Since EIMs may be the prodrome of IBD, we also investigate the chronological relationship between O-EIMs and IBD.

## 2. Methods

This systematic review and meta-analysis of adults with IBD was conducted following the Preferred Reporting Items for Systematic Reviews and Meta-Analyses [13] and Meta-analysis of Observational Studies in Epidemiology [14] guidelines. We focused on observational studies (including case–control, cross-sectional, and cohort studies) to investigate the association between uveitis and IBD.

### 2.1. Literature Search

Several electronic databases, such as Ovid PubMed (503), Ovid Embase/Medline (1800), Web of Science, and Scopus (217), were comprehensively searched from their inception until 25 June 2022, without any language constraints. An experienced medical librarian conducted and executed the search with cooperation from the investigators using controlled vocabulary augmented with the following keywords: “uveitis”, “anterior uveitis”, “posterior uveitis”, “intermediate uveitis”, “pars planitis”, “choroiditis”, “iritis”, “papillitis”, “vitritis”, “iridocyclitis”, “chorioretinitis”, “retinitis”, OR “panuveitis” AND “Crohn”, “ulcerative colitis”, “inflammatory bowel disease”, OR “IBD” AND “cohort”, “population-based”, “nationwide”, “retrospective”, “epidemiology”, “observational”, “case-control”, OR “cross-sectional”. Two authors (JXL and SNC) independently reviewed the titles and abstracts of selected studies to exclude those that did not address the study question of interest based on predetermined inclusion and exclusion criteria. The entire texts of the remaining articles were studied to determine important information, and disagreements on article selection were addressed by consensus. Subsequently, all relevant papers’ reference lists were recursively searched to identify more studies. Lastly, major ophthalmic conference proceedings from 2011 to 2022 were manually searched to identify abstract-only studies.

### 2.2. Study Selection and Data Extraction

The inclusion criteria were as follows: (1) observational studies examining the association between uveitis and IBD, such as cross-sectional, case–control, and cohort studies; (2) human and adult study participants; and (3) case and control groups consisting of patients with and without uveitis, respectively. This study excluded patients aged <18 years or who were enrolled in a study on pediatric O-EIMs. Data from studies that included both adult and pediatric patients with IBD was required to be distinguishable by those groups. Two authors (JXL and SNC) independently examined the search results and determined their eligibility by reviewing the titles and abstracts of the citations. Potentially eligible studies were reviewed, and those meeting the inclusion criteria were included. Disagreements were addressed by consulting with JML. First author, year of publication, country, study design, and quantitative data on the association between uveitis and IBD were retrieved from the included studies. Additionally, mail was sent to the authors requesting additional information if data of interest were missing during data extraction. If the requested additional information was not received within 1 month, the remaining data of interest were not included in the study.

### 2.3. Statistical Analysis

The pooled odds ratio (OR) and 95% confidence interval (CI) were computed by extracting the number of cases of uveitis, iridocyclitis, or iritis in patients with CD or UC. The heterogeneity of the included studies was assessed using Cochran’s Q test and *I*^2^. Cochran’s Q test with a *p*-value of > 0.1 was considered indicative of no heterogeneity, and an *I*^2^ of < 50% was considered indicative of low heterogeneity. The meta-analysis was conducted using the random-effects model. Results are presented as forest plots, and publication bias was detected using funnel plots, the Egger linear regression test, and the Begg and Mazumdar rank correlation test, with significance set to *p* < 0.10 [15]. When a potential publication bias was found, the trim and fill procedure was utilized for correction. This study conducted a subgroup analysis in different regions. All statistical analyses were conducted using the Comprehensive Meta-Analysis Software v3.0 (Biostat, Inc., Englewood, NJ, USA).

## 3. Results

### 3.1. Literature Search Results

Of 2520 publications, 699 duplicates were removed, and the remaining 1821 studies were screened. Among these, 1766 that were not on the topic of interest were excluded. Eligibility was assessed in 55 articles, and 27 did not meet the inclusion criteria, 2 were not in full text, and 5 did not have the required data for analysis. Finally, 21 articles were selected for subsequent analysis. Figure 1 shows the detailed search strategy and study selection processes.

### 3.2. The Characteristics of Included Studies

There were substantial variations among the studies. Some studies included just O-EIMs existing upon IBD diagnosis, while others considered developing O-EIMs during follow-up. Since O-EIMs may precede IBD diagnosis, all studies will be included for meta-analysis. Some studies only reported uveitis, while others included episcleritis or conjunctivitis. Moreover, the definition of uveitis varied from iridocyclitis, choroiditis, and iritis, to retinitis. Uveitis is categorized into four subclasses based on uveal inflammation: anterior, intermediate, posterior, and panuveitis. Therefore, all these forms will be classified into uveitis. This study included 21 studies comprising 190,941 IBD, including 62,874 CD and 128,067 UC. Table 1 summarizes the characteristics of the study populations.

### 3.3. Association of O-EIMs with IBD

Figure 2 shows the forest plot of the prevalence of O-EIMs in patients with CD or UC. Some studies provided information on indeterminate IBD, which was excluded from the meta-analysis because it could not be categorized into CD or UC. The meta-analysis demonstrated an association between uveitis and CD, and adults with CD have a significantly elevated risk of uveitis (pooled OR 1.603, 95% CI 1.254–2.049, *p* < 0.001) (Figure 2A). Cochran’s chi-square test (*p* < 0.001) and *I*^2^ statistic (*I*^2^ = 76.116%) revealed significant heterogeneity among studies. Nonetheless, the prevalence of episcleritis and conjunctivitis had no significant difference between CD and UC (Figure 2B,C).

### 3.4. Stratification with World Regions

To investigate the disparity between world region and race, we divided included studies into Asia, Central Asia, and Europe. Figure 3 presents the subgroup analyses by region of the study population. The pooled analysis of the Asian population revealed non-significantly increased odds of uveitis in patients with CD (pooled OR 1.428, 95% CI 0.612–3.334, *p* = 0.410). Only two studies in Central Asia were involved in the analysis, without significant results, but they presented a trend that uveitis favored in patients with CD over UC (pooled OR 2.468, 95% CI 0.640–9.525, *p* = 0.190). Conversely, European studies indicated a significantly increased prevalence of uveitis in patients with CD (pooled OR 1.683, 95% CI 1.377–2.056, *p* < 0.001). For episcleritis, the frequency of episcleritis did not differ between CD and UC (Figure 2B). Nonetheless, European populations had a significantly higher prevalence of episcleritis in patients with CD than UC (pooled OR 2.401, 95% CI 1.112–5.184, *p* = 0.026) (Figure 3B).

### 3.5. Chronological Analysis

Only two studies showed uveitis before and after IBD diagnosis. The pooled analysis indicated no significant finding (pooled OR 2.152, 95% CI 0.136–33.964, *p* = 0.586) since extreme heterogeneity (*I*^2^ = 99.516%) (Figure 4).

### 3.6. Period Prevalence of O-EIMs

Figure 5 illustrates the overall prevalence of uveitis, episcleritis, and conjunctivitis. The period prevalence of uveitis in CD and UC ranged from 0–21.57% and 0–4.76%, respectively. While the period prevalence of episcleritis in CD and UC was 0–5.41% and 0–24.14%, respectively. 

## 4. Discussion

The present study revealed that patients with CD had an approximately 1.6-fold significantly increased odds of developing uveitis compared with UC. In the subgroup analysis, the European population had significantly elevated odds of uveitis in CD, but not for Central Asia and Asia populations. The overall prevalence of episcleritis was not different between the CD and UC, whereas the odds of episcleritis were significantly higher in CD over UC in European populations.

Among the included studies, some studies described an extremely high prevalence of O-EIMs in CD or UC. Bandyopadhyay et al. [10] revealed a disproportionately higher uveitis prevalence in CD and substantially higher episcleritis prevalence in UC than that reported in cohorts from other countries based on the Indian population, probably due to a selection bias of a cohort with more severe diseases, because half of the included patients with CD and UC were on systemic steroids. These studies reporting outliers were all comprised of a small-size population, which was less than 3000 participants. Therefore, the selection bias effect is prominent. The prevalence of uveitis in CD and UC was 1.66% and 1.26%, respectively, and for episcleritis, it was 0.26% and 0.21%, according to studies with more than 3000 individuals (data not shown). The findings contradict the data from prior investigations, which indicated the prevalence of episcleritis was higher than uveitis in IBD [3]. Overall, the prevalence of ocular inflammation remained low in large cohort studies because, first, a large part of patients with IBD received systemic corticosteroid and/or antitumor necrosis factor (TNF)-α agents, which may decrease ocular inflammation. Second, some O-EIMs were observed as asymptomatic, thereby underestimating the prevalence of uveitis in individuals with IBD.

In the present study, five population-based cohort studies consisted of more than ten thousand individuals. Burisch et al. [16] indicated that the incidence rate of iridocyclitis in patients with CD and UC was 6.7% (95% Cl 3.9–11.5%) and 3.1% (95% Cl 1.9–5.2%), respectively. The incidence rate ratio of iridocyclitis increased to 8.24 and 3.29 in CD and UC, respectively, compared with the control group. Halling et al. [21] revealed that patients with CD and UC had 3.6-fold (95% Cl 2.7–4.7) and 2.4-fold (95% Cl 2.0–2.9) odds of developing uveitis, respectively, compared with controls. The study by Vadstrup et al. [31] revealed increased odds of developing uveitis in patients with CD (OR 2.87, 95% Cl 2.05–4.02) and UC (OR 2.32, 95% Cl 1.79–3.00) compared with controls. Card et al. [17] reported the adjusted odd ratios of uveitis in patients with CD and UC as 3.20 (95% Cl 2.89–3.55) and 2.35 (95% Cl 2.15–2.56), respectively. These studies demonstrated that uveitis favor CD over UC. However, a nationwide study based on the Korean population had the opposite observation. Yang et al. [32] showed the age- and sex-standardized prevalence of iridocyclitis in CD (OR 1.92, 95% Cl 1.56–2.29) and UC (OR 2.23, 95% Cl 2.00–2.47), though there was no significant difference between both. Bernstein et al. [35] also indicated higher odds of uveitis in the UC (OR 8.6, 95% Cl 4.40–17.0) cohort than in the CD (OR 4.5, 95% Cl 2.20–9.40) cohort in females rather than males.

Regarding episcleritis, only two large-scale population-based cohort studies reported episcleritis. Compared with the control, Halling et al. [21] revealed 5.5-fold- and 2.1-fold-increased odds of episcleritis in Denmark patients with CD and UC, respectively. Yang et al. [32] showed the age- and sex-standardized prevalence of episcleritis in CD (OR 1.28, 95% Cl 0.26–2.30) and UC (OR 2.24, 95% Cl 1.43–3.06) based on a Korean population. The disparity may be due to the races of the study populations. 

Patients with O-EIMs were also much more susceptible to other EIMs, such as arthritis, ankylosing spondylitis, erythema nodosum, and pyoderma gangrenosum. The chance of suffering from a concomitant EIM across all individual EIMs increased 4.69-fold and 2.86-fold for CD and UC in uveitis, respectively [36]. EIMs appear more prevalent in children with IBD but with lower overall frequency than in adults, which may account for the shorter duration of IBD development [37]. Additionally, other available data indicates lower O-EIM prevalence in children than in adults. Otherwise, EIMs in children and adults are similar. EIMs, including arthritis, aphthous stomatitis, and uveitis, are more prevalent in CD than in UC [38]. The meta-analysis conducted by Ottaviano et al. [39] reported the overall O-EIMs prevalence (mostly uveitis) at 0.62–1.82% in children with IBD, which is lower than the prevalence of uveitis in adults with IBD of 0.19%–10.09% in our study. Similarly, children with CD are more likely to develop O-EIMs than children with UC and indeterminate IBD (pooled OR 2.70, 95% CI 1.51–4.83) [39].

Burisch et al. [16] described 15 immune-mediated inflammatory diseases, including iridocyclitis, and divided them into onset before and after CD and UC diagnosis. Card et al. [17] also reported uveitis before and after IBD diagnosis. Since EIMs may act as a prodrome of IBD, any occurrence of O-EIMs before and after diagnosis of IBD will be counted. Lo et al. [40] disclosed an association between uveitis and an increased risk of incident CD (OR 1.44, 95%Cl 1.39–1.49) based on a Taiwanese population. Using United Kingdom data, King et al. [41] showed the adjusted hazard ratio of IBD elevated in patients with anterior uveitis and episcleritis/scleritis as 3.39-fold and 1.73-fold, respectively.

All the evaluated EIMs were diagnosed more frequently after IBD, except for psoriasis and multiple sclerosis. The odds of uveitis after the IBD diagnosis were 4.76-fold (95% Cl 4.39–5.16). Moreover, iridocyclitis developed for an average of 2.2 years (95% Cl 1.3–3.4) from IBD onset [16]. However, Park et al. [30] reported that the median time of O-EIMs occurrence concerning IBD diagnosis was within a month. Card et al. [17] found that EIMs were more likely to occur after an IBD diagnosis. The duration between O-EIMs development and IBD remained various; hence, no consistent results were derived from the meta-analysis. Given that both the onset of IBD symptoms and IBD-related complications may be insidious, it is difficult to establish their chronological correlation. A Future large-scale study with rigorous design may solve the discrepancies.

Several observational studies report that the prevalence of O-EIMs remained low [35,42,43], ranging from 0.3–13.0% among IBD patients [44,45,46]. Furthermore, 1.6–5.4% in cases of ulcerative colitis and 3.5–6.8% in cases of Crohn’s disease [47]. Musculoskeletal manifestations, which are considered the most common EIM, occur in 9–53% of patients with IBD [48]. Moreover, 2–34% of IBD patients have reported dermatologic manifestations [49]. Renal and urinary involvement has been documented to occur in 4–23% of IBD patients, manifested primarily as fistulas, urinary calculi, and ureteral obstruction [50]. The prevalence discrepancies between EIMs of organ systems remained obscure.

The association between O-EIMs and IBD is still not fully established. As a putative mechanism connecting systemic diseases with uveitis, a disruption in normal macrophage-mediated autophagy was hypothesized [51]. Even though the eye is an immune-privileged organ, immunological dysregulation can cause significant inflammation and uveitis in autoimmune disorders. Hereditary factors may also play a role. A large retrospective investigation revealed that a family history of IBD seemed to be an independent risk factor for the development of ocular inflammation, despite the absence of bowel disease or known genetic susceptibility [52]. However, no definite correlation was found between O-EIMs and the activity of IBD. O-EIMs may occur during quiescent or active intestinal inflammation periods and precede the IBD diagnosis [53]. Some patients with IBD and colonic involvement had an asymptomatic uveitis recurrence, implying a higher risk for O-EIMs in IBD.

Even though clinical manifestation and risk factors vary, both CD and UC are believed to result from the interaction between environmental factors, genetic susceptibility, intestinal microbiome changes, and dysregulation of the immune system, leading to compromised intestinal mucosa [54,55]. Owing to the disruption of tight junctions and the mucus film covering the epithelial layer, luminal antigens can enter intestinal epithelial cells. Macrophages and dendritic cells switch to an active state upon recognition of commensal microbiota by molecular pattern-recognition receptors. Consequently, multiple signaling pathways were activated, and the transcription of proinflammatory genes was upregulated, leading to an increase in proinflammatory cytokine production and leucocyte recruitment, which perpetuates the cycle of inflammation [56,57]. These proinflammatory mediators may circulate to the eye and result in ocular inflammation. Additionally, ocular involvement may be also caused by the local action of antigen-antibody complexes generated against the intestinal wall vessels and transferred through the circulation.

CD is characterized by transmural inflammation of the intestine and may affect any part of the gastrointestinal tract, from the mouth to the perineal region, whereas UC is restricted to affect the mucosal layer of the colon. Our study revealed that uveitis and episcleritis are more prevalent in patients with CD than those with UC in European populations. However, in the pooled analysis, it is uveitis rather than episcleritis that remained significantly more prevalent in CD than in UC. Uveitis and episcleritis are associated with infection or inflammation, and a higher level of inflammation of CD may lead to a higher incidence of these ocular manifestations, though the sign may be subtle. Furthermore, western lifestyles, such as urbanization, consumption of fast-food, low-fiber and high-meat diets or increased hygiene, may be key risk factors [58]. Other environmental factors may play critical roles in IBD pathogenesis. Sucralose, discovered in 1976, is 600 times as sweet as sucrose. It was widely used after approval in the United States in 1998 and the European Union in 2004. Several studies indicated that it might be the probable cause of the remarkably increased incidence of IBD [59,60,61,62]. Of note, a marked rise in the incidence of CD but not UC was observed in South-Eastern Norway for the last 15 years [63]. We postulated that sucralose might be a contributor to the increased prevalence of ocular symptoms in CD. Further investigation is warranted to determine the observed phenomenon in this study.

### Strengths and Limitations

The primary strengths of this study were the comprehensive investigation of the prevalence of O-EIMs and IBD and the identification of the disparity of it in different races. The chronological correlation of O-EIMs and IBD was also addressed. Notwithstanding, this study was subjected to a few limitations. First, extracting data from these epidemiological studies is difficult due to their fragmented and heterogeneous nature, which limited subgroup analysis with stratification of sex, age group or current medication for IBD. It was also too complicated to evaluate the risk of bias across studies. Moreover, O-EIMs are not defined as an outcome in all literature, making it hard to assess the risk of bias. Second, no cohort studies reported the severity of uveitis and IBD; therefore, their association was not analyzed in terms of disease activity. Third, only one study in Latin America provided data on the association between uveitis and IBD; thus, subgroup analyses for this region are unavailable. Fourth, the relative weight of studies changed between subgroup analyses because the sample sizes of the included research differed. However, the effect was overall consistent. Lastly, data on conjunctivitis were scarce, making it difficult to obtain a reliable pooled analysis result.

## 5. Conclusions

This meta-analysis revealed that CD has an approximately 1.6-fold increased odds of uveitis compared with UC. In the European population, uveitis and episcleritis were favored in patients with CD than those with UC. Most of the EIMs are more prevalent in CD, which may be implicated in the higher scale of systemic inflammation. Consultation with a gastroenterologist is recommended for people with O-EIMs who exhibit bowel symptoms, especially people with CD. Ophthalmologists must consider that ocular signs of IBD may precede the systemic disease, and persistent O-EIMs of unknown etiology require careful evaluation. 

## Figures and Tables

**Figure 1 ijerph-19-15683-f001:**
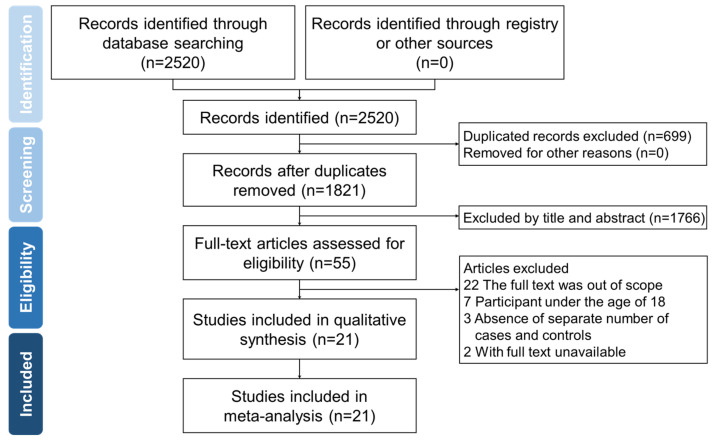
Flow diagram demonstrating the process of study identification.

**Figure 2 ijerph-19-15683-f002:**
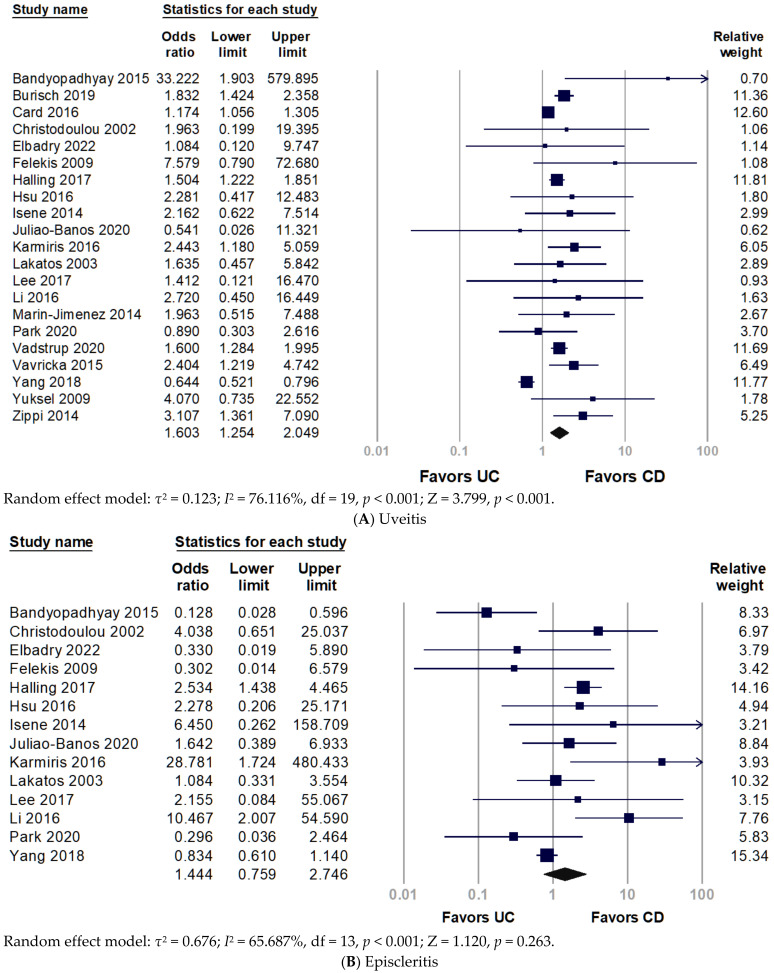
Forest plot for prevalence of ocular extra-intestinal manifestation among adult patients with inflammatory bowel disease: (**A**) uveitis; (**B**) episcleritis; (**C**) conjunctivitis. CD, Crohn’s disease; UC, ulcerative colitis.

**Figure 3 ijerph-19-15683-f003:**
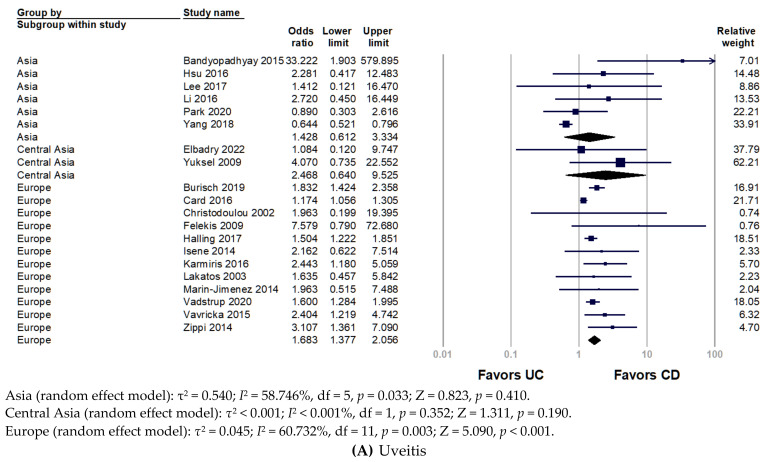
Forest plot for the prevalence of ocular extra-intestinal manifestation among adult patients with inflammatory bowel disease stratified by world region: (**A**) uveitis; (**B**) episcleritis. CD, Crohn’s disease; UC, ulcerative colitis.

**Figure 4 ijerph-19-15683-f004:**
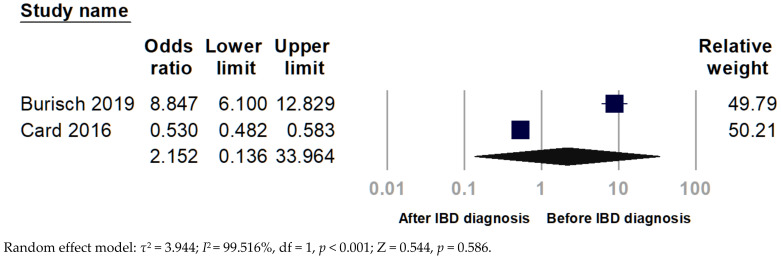
Forest plot for the chronological relationship between uveitis and inflammatory bowel disease (IBD).

**Figure 5 ijerph-19-15683-f005:**
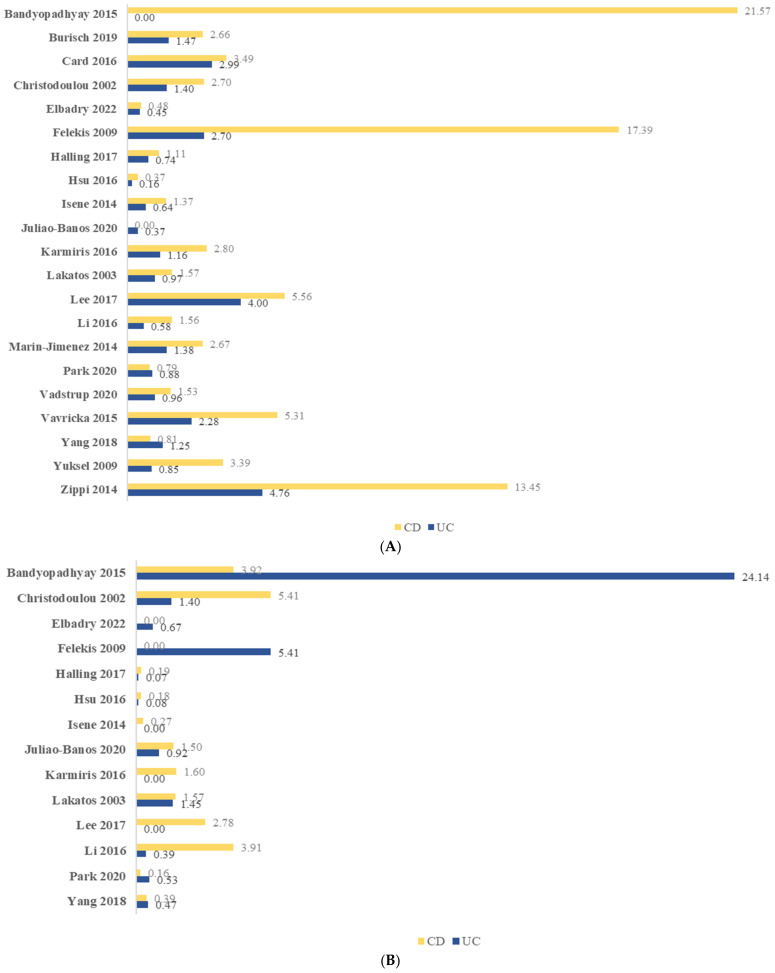
Period prevalence (%) of ocular manifestation among adult patients with inflammatory bowel disease: (**A**) uveitis; (**B**) episcleritis; (**C**) conjunctivitis. CD, Crohn’s disease; UC, ulcerative colitis.

**Table 1 ijerph-19-15683-t001:** Overview of included studies.

Study	Number	Country	Duration (years)	Definition of Uveitis	Uveitis		Episcleritis		Conjunctivitis		IBD	
					CD	UC	CD	UC	CD	UC	CD	UC
Bandyopadhyay 2015 [10]	109	India	1.5	uveitis	11	0	2	14	-	-	51	58
Burisch 2019 [16]	14,377	Denmark	15	iridocyclitis	103	154	-	-	-	-	3879	10,498
Card 2016 [17]	45,312	UK	24	uveitis	636	811	-	-	-	-	18,204	27,108
Christodoulou 2002 [18]	252	Greece	14	iridocyclitis	1	3	2	3	-	-	37	215
Elbadry 2022 [19]	2001	Egypt	3	uveitis	1	4	0	6	-	-	207	897
Felekis 2009 [20]	60	Greece	N/M	uveitis	4	1	0	2	1	0	23	37
Halling 2017 [21]	44,409	Denmark	36	iridocyclitis	148	230	25	23	-	-	13,343	31,066
Hsu 2016 [22]	3153	Taiwan	13	iritis, uveitis	2	4	1	2	-	-	547	2490
Isene 2014 [23]	1145	Multi-national *	2	iridocyclitis, uveitis	5	5	1	0	-	-	364	781
Juliao-Banos 2020 [24]	744	Colombia	N/M	uveitis	0	2	3	5	-	-	200	544
Karmiris 2016 [25]	1860	Greece	N/M	iridocyclitis, vitritis, choroiditis, retinitis	28	10	16	0	-	-	1001	859
Lakatos 2003 [26]	873	Hungary	25	anterior uveitis	4	6	4	9	1	4	254	619
Lee 2017 [27]	61	Korea	1	iritis	2	1	1	0	-	-	36	25
Li 2016 [28]	645	China	20	uveitis	2	3	5	2	-	-	128	517
Marin-Jimenez 2014 [29]	518	Spain	2	uveitis	8	3	-	-	-	-	300	218
Park 2020 [30]	1764	Multi-national ^#^	16	iritis, uveitis	5	10	1	6	-	-	634	1130
Vadstrup 2020 [31]	32,446	Denmark	13	iritis, uveitis	134	199	-	-	-	-	8769	20,722
Vavricka 2015 [4]	1218	Switzerland	4	uveitis	39	11	-	-	-	-	735	483
Yang 2018 [32]	43,281	Korea	1	iridocyclitis	113	368	55	139	-	-	13,925	29,356
Yuksel 2009 [33]	352	Turkey	4.5	uveitis	4	2	-	-	-	-	118	234
Zippi 2014 [34]	329	Italy	12	uveitis	16	10	-	-	-	-	119	210

CD, Crohn’s disease; IBD, inflammatory bowel disease; N/M, not mentioned; UC, ulcerative colitis; UK, United Kingdom. * Nine centers located in Norway, Denmark, Netherlands, Spain, Italy (2 centers), Greece (2 centers), and Israel. Isene’s study was assigned to Europe in subgroup analysis since most centers were located in Europe. ^#^ Korea (3 centers), Malaysia (2 centers), Myanmar (2 centers), Vietnam (2 centers), and Thailand (1 center).

## Data Availability

The data will be provided by the first author upon reasonable request.

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
