# Peer review of "The Prevalence of Ocular Extra-Intestinal Manifestations in Adults Inflammatory Bowel Disease: A Systematic Review and Meta-Analysis"

_ijerph, 2022, doi:10.3390/ijerph192315683_

Round 1
Reviewer 1 Report
In this paper, the authors want to examine the prevalence of O-EIMs in patients with CD and UC. Some significant concerns (see below) deter the enthusiasm.
1.This study examines the prevalence of O-EIMs in patients with CD and UC. They concluded that O-EIMs seemed to be more prevalent in CD than in UC. The title is inappropriate.
2. Different regions have different manifestations of ocular inflammation, and the authors have conducted many studies but without reaching specific conclusions. Other complications related to IBD were not studied but were discussed in focus.
3. How to explain that some data do not match the conclusions should be discussed.
4. Writing and English need revision.
Reviewer 2 Report
I find it is well presented and the topic of revealing ocular manifestations in adults with inflammatory bowel disease is interesting and fit the general scope of the journal.
Author Response
We made every effort to enhance the quality of the entire study. Hope to publish with you.
Round 2
Reviewer 1 Report
After major revision, most concerns were resolved. Low morbidity is still a big question for this study.
Author Response
We added the reponse to the reviewer's comment in the Discission section on page 10. Please refer to the revision.
